# Self-Consistent Dynamical Field Theory of Kernel Evolution in Wide Neural Networks

**Blake Bordelon  &  Cengiz Pehlevan**

John Paulson School of Engineering and Applied Sciences, Center for Brain Science
Harvard University
Cambridge MA, 02138
`blake_bordelon@g.harvard.edu, cpehlevan@g.harvard.edu`

## Abstract

We analyze feature learning in infinite-width neural networks trained with gradient flow through a self-consistent dynamical field theory. We construct a collection of deterministic dynamical order parameters which are inner-product kernels for hidden unit activations and gradients in each layer at pairs of time points, providing a reduced description of network activity through training. These kernel order parameters collectively define the hidden layer activation distribution, the evolution of the neural tangent kernel, and consequently output predictions. We show that the field theory derivation recovers the recursive stochastic process of infinite-width feature learning networks obtained from Yang & Hu with Tensor Programs [1]. For deep linear networks, these kernels satisfy a set of algebraic matrix equations. For nonlinear networks, we provide an alternating sampling procedure to self-consistently solve for the kernel order parameters. We provide comparisons of the self-consistent solution to various approximation schemes including the static NTK approximation, gradient independence assumption, and leading order perturbation theory, showing that each of these approximations can break down in regimes where general self-consistent solutions still provide an accurate description. Lastly, we provide experiments in more realistic settings which demonstrate that the loss and kernel dynamics of CNNs at fixed feature learning strength is preserved across different widths on a CIFAR classification task.

## 1   Introduction

Deep learning has emerged as a successful paradigm for solving challenging machine learning and computational problems across a variety of domains [2, 3]. However, theoretical understanding of the training and generalization of modern deep learning methods lags behind current practice. Ideally, a theory of deep learning would be analytically tractable, efficiently computable, capable of predicting network performance and internal features that the network learns, and interpretable through a reduced description involving desirably initialization-independent quantities.

Several recent theoretical advances have fruitfully considered the idealization of *wide neural networks*, where the number of hidden units in each layer is taken to be large. Under certain parameterization, Bayesian neural networks and gradient descent trained networks converge to gaussian processes (NNGPs) [4–6] and neural tangent kernel (NTK) machines [7–9] in their respective infinite-width limits. These limits provide both analytic tractability as well as detailed training and generalization analysis [10–17]. However, in this limit, with these parameterizations, data representations are fixed and do not adapt to data, termed the *lazy regime* of NN training, to contrast it from the *rich regime* where NNs significantly alter their internal features while fitting the data [18, 19]. The fact that the representation of data is fixed renders these kernel-based theories incapable of explaining feature

learning, an ingredient which is crucial to the success of deep learning in practice [20, 21]. Thus, alternative theories capable of modeling feature learning dynamics are needed.

Recently developed alternative parameterizations such as the mean field [22] and the $\mu P$ [1] parameterizations allow feature learning in infinite-width NNs trained with gradient descent. Using the Tensor Programs framework, Yang & Hu identified a stochastic process that describes the evolution of preactivation features in infinite-width $\mu P$ NNs [1]. In this work, we study an equivalent parameterization to $\mu P$ with self-consistent dynamical mean field theory (DMFT) and recover the stochastic process description of infinite NNs using this alternative technique. In the same large width scaling, we include a scalar parameter $\gamma_0$ that allows smooth interpolation between lazy and rich behavior [18]. We provide a new computational procedure to sample this stochastic process and demonstrate its predictive power for wide NNs.

Our novel contributions in this paper are the following:

1. We develop a path integral formulation of gradient flow dynamics in infinite-width networks in the feature learning regime. Our parameterization includes a scalar parameter $\gamma_0$ to allow interpolation between rich and lazy regimes and comparison to perturbative methods.

2. Using a stationary action argument, we identify a set of saddle point equations that the kernels satisfy at infinite-width, relating the stochastic processes that define hidden activation evolution to the kernels and vice versa. We show that our saddle point equations recover at $\gamma_0 = 1$, from an alternative method, the same stochastic process obtained previously with Tensor Programs [1].

3. We develop a polynomial-time numerical procedure to solve the saddle point equations for deep networks. In numerical experiments, we demonstrate that solutions to these self-consistency equations are predictive of network training at a variety of feature learning strengths, widths and depths. We provide comparisons of our theory to various approximate methods, such as perturbation theory.

## 1.1 Related Works

A natural extension to the lazy NTK/NNGP limit that allows the study of feature learning is to calculate finite width corrections to the infinite-width limit. Finite width corrections to Bayesian inference in wide networks have been obtained with various perturbative [23–29] and self-consistent techniques [30–33]. In the gradient descent based setting, leading order corrections to the NTK dynamics have been analyzed to study finite width effects [34–36, 27]. These methods give approximate corrections which are accurate provided the strength of feature learning is small. In very rich feature learning regimes, however, the leading order corrections can give incorrect predictions [37, 38].

Another approach to study feature learning is to alter NN parameterization in gradient-based learning to allow significant feature evolution even at infinite-width, the *mean field* limit [22, 39]. Works on mean field NNs have yielded formal loss convergence results [40, 41] and shown equivalences of gradient flow dynamics to a partial differential equation (PDE) [42–44].

Our results are most closely related to a set of recent works which studied infinite-width NNs trained with gradient descent (GD) using the Tensor Programs (TP) framework [1]. We show that our discrete time field theory at unit feature learning strength $\gamma_0 = 1$ recovers the stochastic process which was derived from TP. The stochastic process derived from TP has provided insights into practical issues in NN training such as hyper-parameter search [45]. Computing the exact infinite-width limit of GD has exponential time requirements [1], which we show can be circumvented with an alternating sampling procedure. A projected variant of GD training has provided an infinite-width theory that could be scaled to realistic datasets like CIFAR-10 [46]. Inspired by Chizat and Bach's work on mechanisms of lazy and rich training [18], our theory interpolates between lazy and rich behavior in the mean field limit for varying $\gamma_0$ and allows comparison of DMFT to perturbative analysis near small $\gamma_0$. Further, our derivation of a DMFT action allows the possibility of pursuing finite width effects.

Our theory is inspired by self-consistent dynamical mean field theory (DMFT) from statistical physics [47–53]. This framework has been utilized in the theory of random recurrent networks [54–59], tensor PCA [60, 61], phase retrieval [62], and high-dimensional linear classifiers [63–66], but has yet to be developed for deep feature learning. By developing a self-consistent DMFT of deep NNs, we gain insight into how features evolve in the rich regime of network training, while retaining many pleasant analytic properties of the infinite-width limit.

## 2 Problem Setup and Definitions

Our theory applies to infinite-width networks, both fully-connected and convolutional. For notational ease we will relegate convolutional results to later sections. For input $\boldsymbol{x}_\mu \in \mathbb{R}^D$, we define the hidden *pre-activation* vectors $\boldsymbol{h}^\ell \in \mathbb{R}^N$ for layers $\ell \in \{1, ..., L\}$ as

$$f_\mu = \frac{1}{\gamma\sqrt{N}} \boldsymbol{w}^L \cdot \phi(\boldsymbol{h}_\mu^L) \,, \quad \boldsymbol{h}_\mu^{\ell+1} = \frac{1}{\sqrt{N}} \boldsymbol{W}^\ell \phi(\boldsymbol{h}_\mu^\ell) \,, \quad \boldsymbol{h}_\mu^1 = \frac{1}{\sqrt{D}} \boldsymbol{W}^0 \boldsymbol{x}_\mu, \tag{1}$$

where $\boldsymbol{\theta} = \text{Vec}\{\boldsymbol{W}^0, ..., \boldsymbol{w}^L\}$ are the trainable parameters of the network and $\phi$ is a twice differentiable activation function. Inspired by previous works on the mechanisms of lazy gradient based training, the parameter $\gamma$ will control the laziness or richness of the training dynamics [18, 19, 1, 42]. Each of the trainable parameters are initialized as Gaussian random variables with unit variance $W_{ij}^\ell \sim \mathcal{N}(0,1)$. They evolve under gradient flow $\frac{d}{dt}\boldsymbol{\theta} = -\gamma^2 \nabla_{\boldsymbol{\theta}} \mathcal{L}$. The choice of learning rate $\gamma^2$ causes $\frac{d}{dt}\mathcal{L}|_{t=0}$ to be independent of $\gamma$. To characterize the evolution of weights, we introduce backpropagation variables $\boldsymbol{g}_\mu^\ell = \gamma\sqrt{N}\frac{\partial f_\mu}{\partial \boldsymbol{h}_\mu^\ell} = \dot{\phi}(\boldsymbol{h}_\mu^\ell) \odot \boldsymbol{z}_\mu^\ell$, where $\boldsymbol{z}_\mu^\ell = \frac{1}{\sqrt{N}} \boldsymbol{W}^{\ell\top} \boldsymbol{g}_\mu^{\ell+1}$ is the *pre-gradient* signal.

The relevant dynamical objects to characterize feature learning are feature and gradient kernels for each hidden layer $\ell \in \{1, ..., L\}$, defined as

$$\Phi_{\mu\alpha}^\ell(t,s) = \frac{1}{N}\phi(\boldsymbol{h}_\mu^\ell(t)) \cdot \phi(\boldsymbol{h}_\alpha^\ell(s)) \,, \quad G_{\mu\alpha}^\ell(t,s) = \frac{1}{N}\boldsymbol{g}_\mu^\ell(t) \cdot \boldsymbol{g}_\alpha^\ell(s). \tag{2}$$

From the kernels $\{\Phi^\ell, G^\ell\}_{\ell=1}^L$, we can compute the *Neural Tangent Kernel* $K_{\mu\alpha}^{NTK}(t,s) = \nabla_{\boldsymbol{\theta}} f_\mu(t) \cdot \nabla_{\boldsymbol{\theta}} f_\alpha(s) = \sum_{\ell=0}^L G_{\mu\alpha}^{\ell+1}(t,s)\Phi_{\mu\alpha}^\ell(t,s)$, [7] and the dynamics of the network function $f_\mu$

$$\frac{d}{dt}f_\mu(t) = \sum_{\alpha=1}^P K_{\mu\alpha}^{NTK}(t,t)\Delta_\alpha(t) \,, \quad \Delta_\mu(t) = -\frac{\partial}{\partial f_\mu}\mathcal{L}|_{f_\mu(t)}, \tag{3}$$

where we define base cases $G_{\mu\alpha}^{L+1}(t,s) = 1$, $\Phi_{\mu\alpha}^0(t,s) = K_{\mu\alpha}^x = \frac{1}{D}\boldsymbol{x}_\mu \cdot \boldsymbol{x}_\alpha$. In prior work, $\Phi^\ell, G^\ell$ were termed *forward* and *backward* kernels and were theoretically computed at initialization and empirically measured through training [67]. Our DMFT will provide exact formulas for these kernels throughout the full dynamics of feature learning. We note that the above formula holds for any data point $\mu$ which may or may not be in the set of $P$ training examples. The above expressions demonstrate that knowledge of the temporal trajectory of the NTK on the $t = s$ diagonal gives the temporal trajectory of the network predictions $f_\mu(t)$.

Following prior works on infinite-width networks [22, 1, 40, 19], we study the mean field limit

$$N, \gamma \to \infty \,, \quad \gamma_0 = \frac{\gamma}{\sqrt{N}} = \mathcal{O}_N(1) \tag{4}$$

As we demonstrate in the Appendix D and N, this is the only $N$-scaling which allows feature learning as $N \to \infty$. The $\gamma_0 = 0$ limit recovers the static NTK limit [7]. We discuss other scalings and parameterizations in Appendix N, relating our work to the $\mu P$-parameterization and TP analysis of [1], showing they have identical feature dynamics in the infinite-width limit. We also analyze the effect of different hidden layer widths and initialization variances in the Appendix D.8. We focus on equal widths and NTK parameterization (as in eq. (1)) in the main text to reduce complexity.

## 3 Self-consistent DMFT

Next, we derive our self-consistent DMFT in a limit where $t, P = \mathcal{O}_N(1)$. Our goal is to build a description of training dynamics purely based on representations, and independent of weights. Studying feature learning at infinite-width enjoys several analytical properties:

- The kernel order parameters $\Phi^\ell, G^\ell$ concentrate over random initializations but are dynamical, allowing flexible adaptation of features to the task structure.
- In each layer $\ell$, each neuron's preactivation $h_i^\ell$ and pregradient $z_i^\ell$ become i.i.d. draws from a distribution characterized by a set of order parameters $\{\Phi^\ell, G^\ell, A^\ell, B^\ell\}$.
- The kernels are defined as self-consistent averages (denoted by $\langle\rangle$) over this distribution of neurons in each layer $\Phi_{\mu\alpha}^\ell(t,s) = \langle \phi(h_\mu^\ell(t))\phi(h_\alpha^\ell(s)) \rangle$ and $G_{\mu\alpha}^\ell(t,s) = \langle g_\mu^\ell(t)g_\alpha^\ell(s) \rangle$.

The next section derives these facts from a path-integral formulation of gradient flow dynamics.

## 3.1 Path Integral Construction

Gradient flow after a random initialization of weights defines a high dimensional stochastic process over initalizations for variables $\{\boldsymbol{h}, \boldsymbol{g}\}$. Therefore, we will utilize DMFT formalism to obtain a reduced description of network activity during training. For a simplified derivation of the DMFT for the two-layer ($L = 1$) case, see D.2. Generally, we separate the contribution on each forward/backward pass between the initial condition and gradient updates to weight matrix $\boldsymbol{W}^\ell$, defining new stochastic variables $\boldsymbol{\chi}^\ell, \boldsymbol{\xi}^\ell \in \mathbb{R}^N$ as

$$\boldsymbol{\chi}_\mu^{\ell+1}(t) = \frac{1}{\sqrt{N}} \boldsymbol{W}^\ell(0) \phi(\boldsymbol{h}_\mu^\ell(t)), \quad \boldsymbol{\xi}_\mu^\ell(t) = \frac{1}{\sqrt{N}} \boldsymbol{W}^\ell(0)^\top \boldsymbol{g}_\mu^{\ell+1}(t). \tag{5}$$

We let $Z$ represent the moment generating functional (MGF) for these stochastic fields

$$Z[\{\boldsymbol{j}^\ell, \boldsymbol{v}^\ell\}] = \left\langle \exp\left( \sum_{\ell,\mu} \int_0^\infty dt \left[ \boldsymbol{j}_\mu^\ell(t) \cdot \boldsymbol{\chi}_\mu^\ell(t) + \boldsymbol{v}_\mu^\ell(t) \cdot \boldsymbol{\xi}_\mu^\ell(t) \right] \right) \right\rangle_{\{\boldsymbol{W}^0(0),\ldots \boldsymbol{w}^L(0)\}},$$

which requires, by construction the normalization condition $Z[\{\boldsymbol{0}, \boldsymbol{0}\}] = 1$. We enforce our definition of $\boldsymbol{\chi}, \boldsymbol{\xi}$ using an integral representation of the delta-function. Thus for each sample $\mu \in [P]$ and each time $t \in \mathbb{R}_+$, we multiply $Z$ by

$$1 = \int_{\mathbb{R}^N} \int_{\mathbb{R}^N} \frac{d\boldsymbol{\chi}_\mu^{\ell+1}(t) d\hat{\boldsymbol{\chi}}_\mu^{\ell+1}(t)}{(2\pi)^N} \exp\left( i\hat{\boldsymbol{\chi}}_\mu^{\ell+1}(t) \cdot \left[ \boldsymbol{\chi}_\mu^{\ell+1}(t) - \frac{1}{\sqrt{N}} \boldsymbol{W}^\ell(0) \phi(\boldsymbol{h}_\mu^\ell(t)) \right] \right), \tag{6}$$

for $\boldsymbol{\chi}$ and the respective expression for $\boldsymbol{\xi}$. After making such substitutions, we perform integration over initial Gaussian weight matrices to arrive at an integral expression for $Z$, which we derive in the appendix D.4. We show that $Z$ can be described by set of order-parameters $\{\Phi^\ell, \hat{\Phi}^\ell, G^\ell, \hat{G}^\ell, A^\ell, B^\ell\}$

$$Z[\{\boldsymbol{j}^\ell, \boldsymbol{v}^\ell\}] \propto \int \prod_{\ell\mu\alpha ts} d\Phi_{\mu\alpha}^\ell(t,s) d\hat{\Phi}_{\mu\alpha}^\ell(t,s) dG_{\mu\alpha}^\ell(t,s) d\hat{G}_{\mu\alpha}^\ell(t,s) dA_{\mu\alpha}^\ell(t,s) dB_{\mu\alpha}^\ell(t,s) \tag{7}$$

$$\times \exp\left( NS[\{\Phi, \hat{\Phi}, G, \hat{G}, A, B, j, v\}] \right),$$

$$S = \sum_{\ell\mu\alpha} \int_0^\infty dt \int_0^\infty ds \left[ \Phi_{\mu\alpha}^\ell(t,s) \hat{\Phi}_{\mu\alpha}^\ell(t,s) + G_{\mu\alpha}^\ell(t,s) \hat{G}_{\mu\alpha}^\ell(t,s) - A_{\mu\alpha}^\ell(t,s) B_{\mu\alpha}^\ell(t,s) \right]$$

$$+ \ln \mathcal{Z}[\{\Phi, \hat{\Phi}, G, \hat{G}, A, B, j, v\}], \tag{8}$$

where $S$ is the DMFT action and $\mathcal{Z}$ is a single-site MGF, which defines the distribution of fields $\{\chi^\ell, \xi^\ell\}$ over the neural population in each layer. The kernels $A$ and $B$ are related to the correlations between feedforward and feedback signals in the network. We provide a detailed formula for $\mathcal{Z}$ in the Appendix D.4 and show that it factorizes over different layers $\mathcal{Z} = \prod_{\ell=1}^L \mathcal{Z}_\ell$.

## 3.2 Deriving the DMFT Equations from the Path Integral Saddle Point

As $N \to \infty$, the moment-generating function $Z$ is exponentially dominated by the saddle point of $S$. The equations that define this saddle point also define our DMFT. We thus identify the kernels that render $S$ locally stationary ($\delta S = 0$). The most important equations are those which define $\{\Phi^\ell, G^\ell\}$

$$\frac{\delta S}{\delta \hat{\Phi}_{\mu\alpha}^\ell(t,s)} = \Phi_{\mu\alpha}^\ell(t,s) + \frac{1}{\mathcal{Z}} \frac{\delta \mathcal{Z}}{\delta \hat{\Phi}_{\mu\alpha}^\ell(t,s)} = \Phi_{\mu\alpha}^\ell(t,s) - \left\langle \phi(h_\mu^\ell(t)) \phi(h_\alpha^\ell(s)) \right\rangle = 0,$$

$$\frac{\delta S}{\delta \hat{G}_{\mu\alpha}^\ell(t,s)} = G_{\mu\alpha}^\ell(t,s) + \frac{1}{\mathcal{Z}} \frac{\delta \mathcal{Z}}{\delta \hat{G}_{\mu\alpha}^\ell(t,s)} = G_{\mu\alpha}^\ell(t,s) - \left\langle g_\mu^\ell(t) g_\alpha^\ell(s) \right\rangle = 0, \tag{9}$$

where $\langle\rangle$ denotes an average over the stochastic process induced by $\mathcal{Z}$, which is defined below

$$\{u_\mu^\ell(t)\}_{\mu\in[P],t\in\mathbb{R}_+} \sim \mathcal{GP}(0, \boldsymbol{\Phi}^{\ell-1}), \quad \{r_\mu^\ell(t)\}_{\mu\in[P],t\in\mathbb{R}_+} \sim \mathcal{GP}(0, \boldsymbol{G}^{\ell+1}),$$

$$h_\mu^\ell(t) = u_\mu^\ell(t) + \gamma_0 \int_0^t ds \sum_{\alpha=1}^P \left[ A_{\mu\alpha}^{\ell-1}(t,s) + \Delta_\alpha(s) \Phi_{\mu\alpha}^{\ell-1}(t,s) \right] z_\alpha^\ell(s) \dot{\phi}(h_\alpha^\ell(s)),$$

$$z_\mu^\ell(t) = r_\mu^\ell(t) + \gamma_0 \int_0^t ds \sum_{\alpha=1}^P \left[ B_{\mu\alpha}^\ell(t,s) + \Delta_\alpha(s) G_{\mu\alpha}^{\ell+1}(t,s) \right] \phi(h_\alpha^\ell(s)), \tag{10}$$

where we define base cases $\Phi^0_{\mu\alpha}(t,s) = K^x_{\mu\alpha}$ and $G^{L+1}_{\mu\alpha}(t,s) = 1$, $A^0 = B^L = 0$. We see that the fields $\{h^\ell, z^\ell\}$, which represent the single site preactivations and pre-gradients, are implicit functionals of the mean-zero Gaussian processes $\{u^\ell, r^\ell\}$ which have covariances $\langle u^\ell_\mu(t) u^\ell_\alpha(s)\rangle = \Phi^{\ell-1}_{\mu\alpha}(t,s)$ and $\langle r^\ell_\mu(t) r^\ell_\alpha(s)\rangle = G^{\ell+1}_{\mu\alpha}(t,s)$. The other saddle point equations give $A^\ell_{\mu\alpha}(t,s) = \gamma_0^{-1} \left\langle \frac{\delta\phi(h^\ell_\mu(t))}{\delta r^\ell_\alpha(s)} \right\rangle, B^\ell_{\mu\alpha}(t,s) = \gamma_0^{-1} \left\langle \frac{\delta g^{\ell+1}_\mu(t)}{\delta u^{\ell+1}_\alpha(s)} \right\rangle$ which arise due to coupling between the feedforward and feedback signals. We note that, in the lazy limit $\gamma_0 \to 0$, the fields approach Gaussian processes $h^\ell \to u^\ell$, $z^\ell \to r^\ell$. Lastly, the final saddle point equations $\frac{\delta S}{\delta \Phi^\ell} = 0, \frac{\delta S}{\delta G^\ell} = 0$ imply that $\hat{\Phi}^\ell = \hat{G}^\ell = 0$. The full set of equations that define the DMFT are given in D.7.

This theory is easily extended to more general architectures such as networks with varying widths by layer (App. D.8), trainable bias parameter (App. H), multiple (but $\mathcal{O}_N(1)$) output channels (App. I), convolutional architectures (App. G), networks trained with weight decay (App. J), Langevin sampling (App. K) and momentum (App. L), discrete time training (App. M). In Appendix N, we discuss parameterizations which give equivalent feature and predictor dynamics and show our derived stochastic process is equivalent to the $\mu P$ scheme of Yang & Hu [1].

## 4    Solving the Self-Consistent DMFT

The saddle point equations obtained from the field theory discussed in the previous section must be solved self-consistently. By this we mean that, given knowledge of the kernels, we can characterize the distribution of $\{h^\ell, z^\ell\}$, and given the distribution of $\{h^\ell, z^\ell\}$, we can compute the kernels [68, 64]. In the Appendix B, we provide Algorithm 1, a numerical procedure based on this idea to efficiently solve for the kernels with an alternating Monte-Carlo strategy. The output of the algorithm are the dynamical kernels $\Phi^\ell_{\mu\alpha}(t,s), G^\ell_{\mu\alpha}(t,s), A^\ell_{\mu\alpha}(t,s), B^\ell_{\mu\alpha}(t,s)$, from which any network observable can be computed as we discuss in Appendix D. We provide an example of the solution to the saddle point equations compared to training a finite NN in Figure 1. We plot $\Phi^\ell, G^\ell$ at the end of training and the sample-trace of these kernels through time. Additionally, we compare the kernels of finite width $N$ network to the DMFT predicted kernels using a cosine-similarity alignment metric $A(\boldsymbol{\Phi}^{DMFT}, \boldsymbol{\Phi}^{NN}) = \frac{\text{Tr } \boldsymbol{\Phi}^{DMFT}\boldsymbol{\Phi}^{NN}}{|\boldsymbol{\Phi}^{DMFT}||\boldsymbol{\Phi}^{NN}|}$. Additional examples are in Appendix Figures 6 and Figure 7.

### 4.1    Deep Linear Networks: Closed Form Self-Consistent Equations

Deep linear networks ($\phi(h) = h$) are of theoretical interest since they are simpler to analyze than nonlinear networks but preserve non-trivial training dynamics and feature learning [69–73, 25, 32, 23]. In a deep linear network, we can simplify our saddle point equations to algebraic formulas that close in terms of the kernels $H^\ell_{\mu\alpha}(t,s) = \langle h^\ell_\mu(t) h^\ell_\alpha(s)\rangle, G^\ell(t,s) = \langle g^\ell(t) g^\ell(s)\rangle$ [1]. This is a significant simplification since it allows solution of the saddle point equations without a sampling procedure.

To describe the result, we first introduce a vectorization notation $\boldsymbol{h}^\ell = \text{Vec}\{h^\ell_\mu(t)\}_{\mu\in[P],t\in\mathbb{R}_+}$. Likewise we convert kernels $\boldsymbol{H}^\ell = \text{Mat}\{H^\ell_{\mu\alpha}(t,s)\}_{\mu,\alpha\in[P],t,s\in\mathbb{R}_+}$ into matrices. The inner product under this vectorization is defined as $\boldsymbol{a} \cdot \boldsymbol{b} = \int_0^\infty dt \sum_{\mu=1}^P a_\mu(t) b_\mu(t)$. In a practical computational implementation, the theory would be evaluated on a grid of $T$ time points with discrete time gradient descent, so these kernels $\boldsymbol{H}^\ell \in \mathbb{R}^{PT\times PT}$ would indeed be matrices of the appropriate size. The fields $\boldsymbol{h}^\ell, \boldsymbol{g}^\ell$ are linear functionals of independent Gaussian processes $\boldsymbol{u}^\ell, \boldsymbol{r}^\ell$, giving $(\mathbf{I} - \gamma_0^2 \boldsymbol{C}^\ell \boldsymbol{D}^\ell)\boldsymbol{h}^\ell = \boldsymbol{u}^\ell + \gamma_0 \boldsymbol{C}^\ell \boldsymbol{r}^\ell$, $(\mathbf{I} - \gamma_0^2 \boldsymbol{D}^\ell \boldsymbol{C}^\ell)\boldsymbol{g}^\ell = \boldsymbol{r}^\ell + \gamma_0 \boldsymbol{D}^\ell \boldsymbol{u}^\ell$. The matrices $\boldsymbol{C}^\ell$ and $\boldsymbol{D}^\ell$ are causal integral operators which depend on $\{A^{\ell-1}, H^{\ell-1}\}$ and $\{B^\ell, G^{\ell+1}\}$ respectively which we define in Appendix F. The saddle point equations which define the kernels are

$$\boldsymbol{H}^\ell = \langle \boldsymbol{h}^\ell \boldsymbol{h}^{\ell\top}\rangle = (\mathbf{I} - \gamma_0^2 \boldsymbol{C}^\ell \boldsymbol{D}^\ell)^{-1}[\boldsymbol{H}^{\ell-1} + \gamma_0^2 \boldsymbol{C}^\ell \boldsymbol{G}^{\ell+1}\boldsymbol{C}^{\ell\top}]\left[(\mathbf{I} - \gamma_0^2 \boldsymbol{C}^\ell \boldsymbol{D}^\ell)^{-1}\right]^\top$$

$$\boldsymbol{G}^\ell = \langle \boldsymbol{g}^\ell \boldsymbol{g}^{\ell\top}\rangle = \left(\mathbf{I} - \gamma_0^2 \boldsymbol{D}^\ell \boldsymbol{C}^\ell\right)^{-1}\left[\boldsymbol{G}^{\ell+1} + \gamma_0^2 \boldsymbol{D}^\ell \boldsymbol{H}^{\ell-1}\boldsymbol{D}^{\ell\top}\right]\left[\left(\mathbf{I} - \gamma_0^2 \boldsymbol{D}^\ell \boldsymbol{C}^\ell\right)^{-1}\right]^\top. \quad (11)$$

Examples of the predictions obtained by solving these systems of equations are provided in Figure 2. We see that these DMFT equations describe kernel evolution for networks of a variety of depths and that the change in each layer's kernel increases with the depth of the network.

Unlike many prior results [69–72], our DMFT does not require any restrictions on the structure of the input data but hold for any $\boldsymbol{K}^x, \boldsymbol{y}$. However, for whitened data $\boldsymbol{K}^x = \mathbf{I}$ we show in

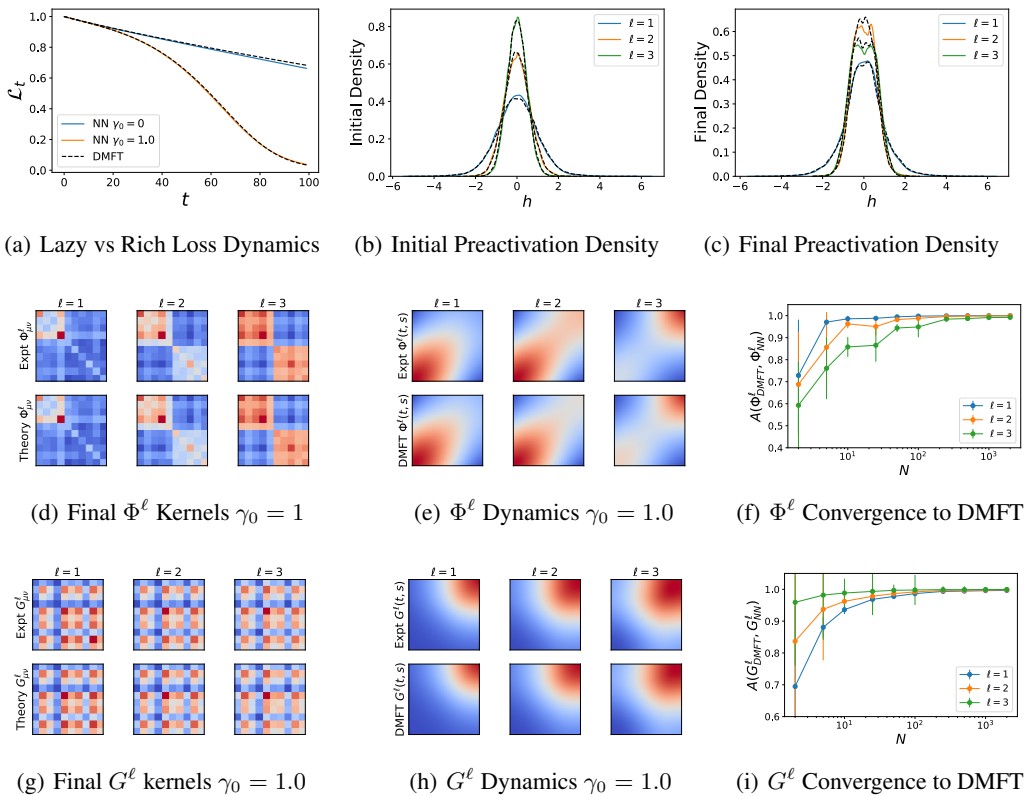

(a) Lazy vs Rich Loss Dynamics    (b) Initial Preactivation Density    (c) Final Preactivation Density

(d) Final $\Phi^\ell$ Kernels $\gamma_0 = 1$    (e) $\Phi^\ell$ Dynamics $\gamma_0 = 1.0$    (f) $\Phi^\ell$ Convergence to DMFT

(g) Final $G^\ell$ kernels $\gamma_0 = 1.0$    (h) $G^\ell$ Dynamics $\gamma_0 = 1.0$    (i) $G^\ell$ Convergence to DMFT

Figure 1: Neural network feature learning dynamics is captured by self-consistent dynamical mean field theory (DMFT). (a) Training loss curves on a subsample of $P = 10$ CIFAR-10 training points in a depth 4 ($L = 3$, $N = 2500$) tanh network ($\phi(h) = \tanh(h)$) trained with MSE. Increasing $\gamma_0$ accelerates training. (b)-(c) The distribution of preactivations at the beginning and end of training matches predictions of the DMFT. (d) The final $\Phi^\ell$ (at $t = 100$) kernel order parameters match the finite width network. (e) The temporal dynamics of the sample-traced kernels $\sum_\mu \Phi^\ell_{\mu\mu}(t, s)$ matches experiment and reveals rich dynamics across layers. (f) The alignment $A(\mathbf{\Phi}^\ell_{DMFT}, \mathbf{\Phi}^\ell_{NN})$, defined as cosine similarity, of the kernel $\Phi^\ell_{\mu\alpha}(t, s)$ predicted by theory (DMFT) and width $N$ networks for different $N$ but fixed $\gamma_0 = \gamma/\sqrt{N}$. Errorbars show standard deviation computed over 10 repeats. Around $N \sim 500$ DMFT begins to show near perfect agreement with the NN. (g)-(i) The same plots but for the gradient kernel $\mathbf{G}^\ell$. Whereas finite width effects for $\mathbf{\Phi}^\ell$ are larger at later layers $\ell$ since variance accumulates on the forward pass, fluctuations in $\mathbf{G}^\ell$ are large in early layers.

Appendix F.1.1, F.2 that our DMFT learning curves interpolate between NTK dynamics and the sigmoidal trajectories of prior works [69, 70] as $\gamma_0$ is increased. For example, in the two layer ($L = 1$) linear network with $\mathbf{K}^x = \mathbf{I}$, the dynamics of the error norm $\Delta(t) = ||\mathbf{\Delta}(t)||$ takes the form $\frac{\partial}{\partial t}\Delta(t) = -2\sqrt{1 + \gamma_0^2(y - \Delta(t))^2}\Delta(t)$ where $y = ||\mathbf{y}||$. These dynamics give the linear convergence rate of the NTK if $\gamma_0 \to 0$ but approaches logistic dynamics of [70] as $\gamma_0 \to \infty$. Further, $\mathbf{H}(t) = \langle \mathbf{h}^1(t)\mathbf{h}^1(t)^\top \rangle \in \mathbb{R}^{P \times P}$ only grows in the $\mathbf{y}\mathbf{y}^\top$ direction with $H_y(t) = \frac{1}{y^2}\mathbf{y}^\top \mathbf{H}(t)\mathbf{y} = \sqrt{1 + \gamma_0^2(y - \Delta(t))^2}$. At the end of training $\mathbf{H}(t) \to \mathbf{I} + \frac{1}{y^2}[\sqrt{1 + \gamma_0^2 y^2} - 1]\mathbf{y}\mathbf{y}^\top$, recovering the rank one spike which was recently obtained in the small initialization limit [74]. We show this one dimensional system in Figure 8.

## 4.2 Feature Learning with L2 Regularization

As we show in Appendix J, the DMFT can be extended to networks trained with weight decay $\frac{d\boldsymbol{\theta}}{dt} = -\gamma^2 \nabla_{\boldsymbol{\theta}} \mathcal{L} - \lambda \boldsymbol{\theta}$. If neural network is homogenous in its parameters so that $f(c\boldsymbol{\theta}) = c^\kappa f(\boldsymbol{\theta})$ (examples include networks with linear, ReLU, quadratic activations), then the final network predictor

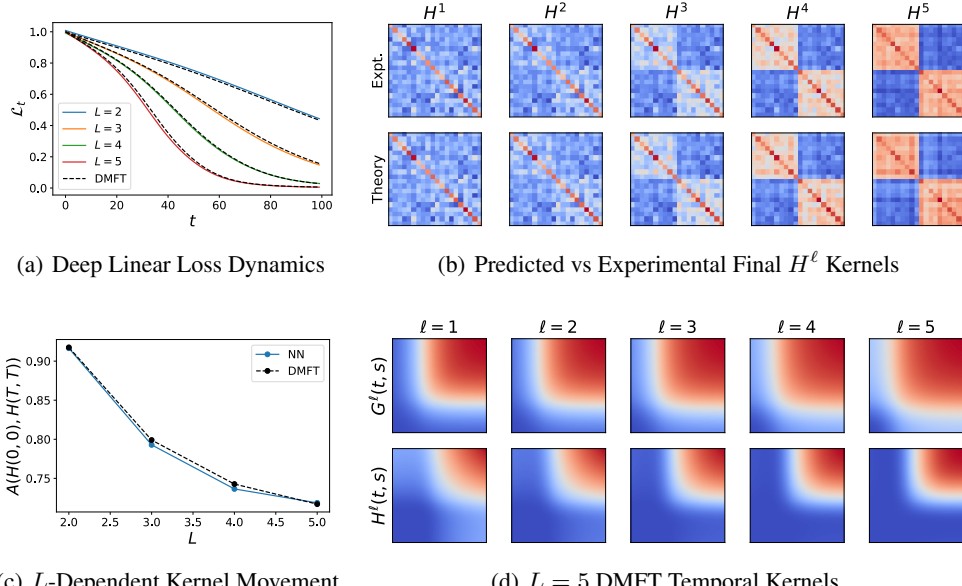

(a) Deep Linear Loss Dynamics

(b) Predicted vs Experimental Final $H^\ell$ Kernels

(c) $L$-Dependent Kernel Movement

(d) $L = 5$ DMFT Temporal Kernels

Figure 2: Deep linear network with the full DMFT. (a) The train loss for NNs of varying $L$. (b) For a $L = 5, N = 1000$ NN, the kernels $H^\ell$ at the end of training compared to DMFT theory on $P = 20$ datapoints. (c) The average displacement of feature kernels for different depth networks at same $\gamma_0$ value. For equal values of $\gamma_0$, deeper networks exhibit larger changes to their features, manifested in lower alignment with their initial $t = 0$ kernels $\boldsymbol{H}$. (d) The solution to the temporal components of the $G^\ell(t, s)$ and $\sum_\mu H^\ell_{\mu\mu}(t, s)$ kernels obtained from the self-consistent equations.

is a kernel regressor with the final NTK $\lim_{t\to\infty} f(\boldsymbol{x}, t) = \boldsymbol{k}(\boldsymbol{x})^\top [\boldsymbol{K} + \lambda\kappa\mathbf{I}]^{-1}\boldsymbol{y}$ where $K(\boldsymbol{x}, \boldsymbol{x}')$ is the *final*-NTK, $[\boldsymbol{k}(\boldsymbol{x})]_\mu = K(\boldsymbol{x}, \boldsymbol{x}_\mu)$ and $[\boldsymbol{K}]_{\mu\alpha} = K(\boldsymbol{x}_\mu, \boldsymbol{x}_\alpha)$. We note that the effective regularization $\lambda\kappa$ increases with depth $L$. In NTK parameterization, weight decay in infinite width homogenous networks gives a trivial fixed point $K(\boldsymbol{x}, \boldsymbol{x}') \to 0$ and consequently a zero predictor $f \to 0$ [75]. However, as we show in Figure 3, increasing feature learning $\gamma_0$ can prevent convergence to the trivial fixed point, allowing a non-zero fixed point for $K, f$ even at infinite width. The kernel and function dynamics can be predicted with DMFT. The fixed point is a nontrivial function of the hyperparameters $\lambda, \kappa, L, \gamma_0$.

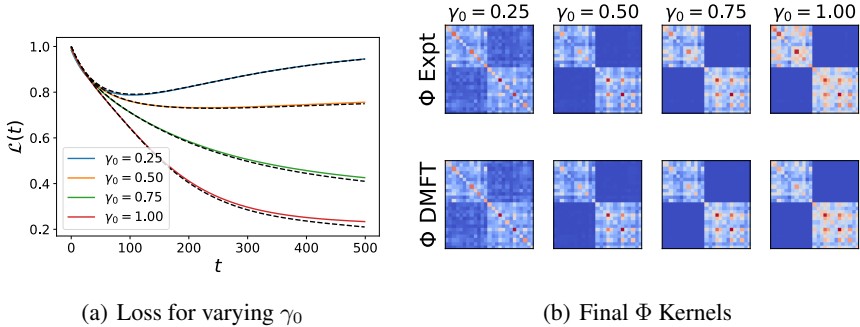

(a) Loss for varying $\gamma_0$

(b) Final $\Phi$ Kernels

Figure 3: Width $N = 1000$ ReLU networks trained with L2 regularization have nontrivial fixed point in DMFT limit ($\gamma_0 > 0$). (a) Training loss dynamics for a $L = 1$ ReLU network with $\lambda = 1$. In $\gamma_0 \to 0$ limit the fixed point is trivial $f = K = 0$. The final loss is a decreasing function of $\gamma_0$. (b) The final kernel is more aligned with target with increasing $\gamma_0$. Networks with homogenous activations enjoy a representer theorem at infinite-width as we show in Appendix J.

# 5 Approximation Schemes

We now compare our exact DMFT with approximations of prior works, providing an explanation of when these approximations give accurate predictions and when they break down.

## 5.1 Gradient Independence Ansatz

We can study the accuracy of the ansatz $\boldsymbol{A}^\ell = \boldsymbol{B}^\ell = 0$, which is equivalent to treating the weight matrices $\boldsymbol{W}^\ell(0)$ and $\boldsymbol{W}^\ell(0)^\top$ which appear in forward and backward passes respectively as independent Gaussian matrices. This assumption was utilized in prior works on signal propagation in deep networks in the lazy regime [76–80]. A consequence of this approximation is the Gaussianity and statistical independence of $\chi^\ell$ and $\xi^\ell$ (conditional on $\{\boldsymbol{\Phi}^\ell, \boldsymbol{G}^\ell\}$) in each layer as we show in Appendix O. This ansatz works very well near $\gamma_0 \approx 0$ (the static kernel regime) since $\frac{d\boldsymbol{h}}{d\boldsymbol{r}}, \frac{d\boldsymbol{z}}{d\boldsymbol{u}} \sim \mathcal{O}(\gamma_0)$ or around initialization $t \approx 0$ but begins to fail at larger values of $\gamma_0, t$ (Figure 4).

## 5.2 Perturbation theory in $\gamma_0$ at infinite-width

In the $\gamma_0 \to 0$ limit, we recover static kernels, giving linear dynamics identical to the NTK limit [7]. Corrections to this lazy limit can be extracted at small but finite $\gamma_0$. This is conceptually similar to recent works which consider perturbation series for the NTK in powers of $1/N$ [35, 27, 28] (though not identical, see Appendix P.7 for finite $N$ effects). We expand all observables $q(\gamma_0)$ in a power series in $\gamma_0$, giving $q(\gamma_0) = q^{(0)} + \gamma_0 q^{(1)} + \gamma_0^2 q^{(2)} + ...$ and compute corrections up to $\mathcal{O}(\gamma_0^2)$. We show that the $\mathcal{O}(\gamma_0)$ and $\mathcal{O}(\gamma_0^3)$ corrections to kernels vanish, giving leading order expansions of the form $\boldsymbol{\Phi} = \boldsymbol{\Phi}^0 + \gamma_0^2 \boldsymbol{\Phi}^2 + \mathcal{O}(\gamma_0^4)$ and $\boldsymbol{G} = \boldsymbol{G}^0 + \gamma_0^2 \boldsymbol{G}^2 + \mathcal{O}(\gamma_0^4)$ (see Appendix P.2). Further, we show that the NTK has relative change at leading order which scales linearly with depth $|\Delta K^{NTK}|/|K^{NTK,0}| \sim \mathcal{O}_{\gamma_0,L}(L\gamma_0^2) = \mathcal{O}_{N,\gamma,L}(\frac{\gamma^2 L}{N})$, which is consistent with finite width effective field theory at $\gamma = \mathcal{O}_N(1)$ [26–28] (Appendix P.6). Further, at the leading order correction, all temporal dependencies are controlled by $P(P+1)$ functions $v_\alpha(t) = \int_0^t ds \Delta_\alpha^0(s)$ and $v_{\alpha\beta}(t) = \int_0^t ds \Delta_\alpha^0(s) \int_0^s ds' \Delta_\beta^0(s')$, which is consistent with those derived for finite width NNs using a truncation of the Neural Tangent Hierarchy [34, 35, 27]. To lighten notation, we focus our main text comparison of our non-perturbative DMFT to perturbation theory in the deep linear case. Full perturbation theory is in Appendix P.2.

Using the timescales derived in the previous section, we find that the leading order correction to the kernels in infinite-width deep linear network have the form

$$K_{\mu\nu}^{NTK}(t, s) = (L+1)K_{\mu\nu}^x + \gamma_0^2 \frac{L(L+1)}{2}K_{\mu\nu}^x \sum_{\alpha\beta} K_{\alpha\beta}^x [v_{\alpha\beta}(t) + v_{\beta\alpha}(s) + v_\alpha(t)v_\beta(s)]$$

$$+ \gamma_0^2 \frac{L(L+1)}{2} \left[ \sum_{\alpha\beta} K_{\mu\alpha}^x K_{\nu\beta}^x [v_{\alpha\beta}(t) + v_{\beta\alpha}(s)] + \sum_{\alpha\beta} K_{\mu\alpha}^x K_{\nu\beta}^x v_\alpha(t)v_\beta(s) \right] + \mathcal{O}(\gamma_0^4). \quad (12)$$

We see that the relative change in the NTK $|\boldsymbol{K}^{NTK} - \boldsymbol{K}^{NTK}(0)|/|\boldsymbol{K}^{NTK}(0)| \sim \mathcal{O}(\gamma_0^2 L) = \mathcal{O}(\gamma^2 L/N)$, so that large depth $L$ networks exhibit more significant kernel evolution, which agrees with other perturbative studies [35, 27, 25] as well as the non-perturbative results in Figure 2. However at large $\gamma_0$ and large $L$, this theory begins to break down as we show in Figure 4.

The DMFT formalism can also be used to extract leading corrections to observables at large but finite width $N$ as we explore in P.7. When deviating from infinite width, the kernels are no longer deterministic over network initializations. The key observation is that the DMFT action $S$ defines a Gibbs measure over the space of kernel order parameters $\boldsymbol{k} = \text{Vec}\{\boldsymbol{\Phi}^\ell, \boldsymbol{G}^\ell, \boldsymbol{A}^\ell, \boldsymbol{B}^\ell\}$ with probability density $\frac{1}{Z} \exp(NS[\boldsymbol{k}])$ where $Z$ is a normalization constant. Near infinite width, any observable average $\langle O(\boldsymbol{k}) \rangle = \frac{1}{Z} \int d\boldsymbol{k} \exp(NS[\boldsymbol{k}]) O(\boldsymbol{k})$ is dominated by order parameters within a $\frac{1}{\sqrt{N}}$ neighborhood of $\boldsymbol{k}^*$. As a consequence, a perturbative series for $\langle O(\boldsymbol{k}) \rangle$ can be obtained from simple averages over Gaussian fluctuations in the kernels $\boldsymbol{k} \sim \mathcal{N}(\boldsymbol{k}^*, -\frac{1}{N}[\nabla^2 S[\boldsymbol{k}^*]]^{-1})$ [29]. The components for $\nabla^2 S[\boldsymbol{k}^*]$ include four point correlations of fields computed over the DMFT distribution.

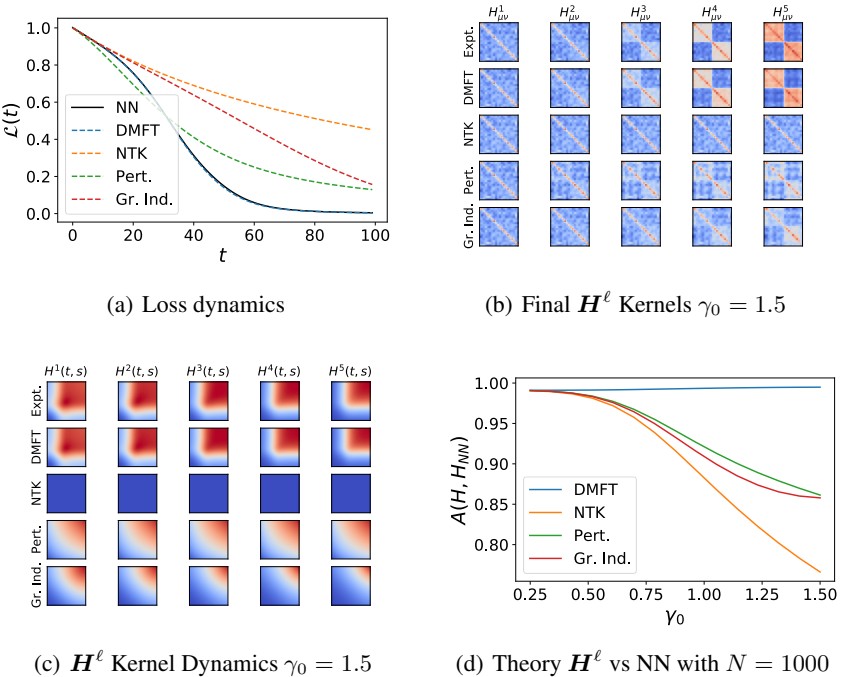

(a) Loss dynamics

(b) Final $\boldsymbol{H}^\ell$ Kernels $\gamma_0 = 1.5$

(c) $\boldsymbol{H}^\ell$ Kernel Dynamics $\gamma_0 = 1.5$

(d) Theory $\boldsymbol{H}^\ell$ vs NN with $N = 1000$

Figure 4: Comparison of DMFT to various approximation schemes in a $L = 5$ hidden layer, width $N = 1000$ linear network with $\gamma_0 = 1.0$ and $P = 100$. (a) The loss for the various approximations do not track the true trajectory induced by gradient descent in the large $\gamma_0$ regime. (b)-(c) The feature kernels $H^\ell_{\mu\alpha}(t, s)$ across each of the $L = 5$ hidden layers for each of the theories is compared to a width 1000 neural network. Again, we plot the sample-traced dynamics $\sum_{\mu\mu} H^\ell_{\mu\mu}(t, s)$. (d) The alignment of $\boldsymbol{H}^\ell$ compared to the finite NN $A(\boldsymbol{H}^\ell, \boldsymbol{H}^\ell_{NN})$ averaged across $\ell \in \{1, ..., 5\}$ for varying $\gamma$. The predictions of all of these theories coincide in the $\gamma_0 = 0$ limit but begin to deviate in the feature learning regime. Only the non-perturbative DMFT is accurate over a wide range of $\gamma_0$.

## 6 Feature Learning Dynamics is Preserved at Fixed $\gamma_0$

Our DMFT suggests that for networks sufficiently wide for their kernels to concentrate, the dynamics of loss and kernels should be invariant under the rescaling $N \to RN, \gamma \to \gamma/\sqrt{R}$, which keeps $\gamma_0$ fixed. To evaluate how well this idea holds in a realistic deep learning problem, we trained CNNs of varying channel counts $N$ on two-class CIFAR classification [81]. We tracked the dynamics of the loss and the last layer $\Phi^L$ kernel. The results are provided in Figure 5. We see that dynamics are largely independent of rescaling as predicted. Further, as expected, larger $\gamma_0$ leads to larger changes in kernel norm and faster alignment to the target function $y$, as was also found in [82]. Consequently, the higher $\gamma_0$ networks train more rapidly. The trend is consistent for width $N = 250$ and $N = 500$. More details about the experiment can be found in Appendix C.2.

## 7 Discussion

We provided a unifying DMFT derivation of feature dynamics in infinite networks trained with gradient based optimization. Our theory interpolates between lazy infinite-width behavior of a static NTK in $\gamma_0 \to 0$ and rich feature learning. At $\gamma_0 = 1$, our DMFT construction agrees with the stochastic process derived previously with the Tensor Programs framework [1]. Our saddle point equations give self-consistency conditions which relate the stochastic fields to the kernels. These equations are exactly solveable in deep linear networks and can be efficiently solved with a numerical method in the nonlinear case. Comparisons with other approximation schemes show that DMFT can be accurate at a much wider range of $\gamma_0$. We believe our framework could be a useful perspective for future theoretical analyses of feature learning and generalization in wide networks.

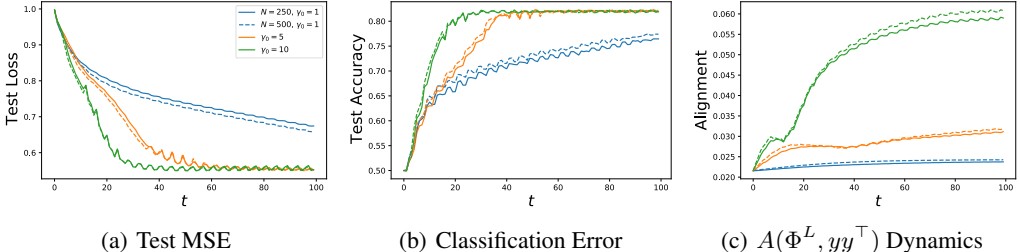

| (a) Test MSE | (b) Classification Error | (c) $A(\Phi^L, yy^\top)$ Dynamics |

Figure 5: The dynamics of a depth 5 ($L = 4$ hidden) CNNs trained on first two classes of CIFAR (boat vs plane) exhibit consistency for different channel counts $N \in \{250, 500\}$ for fixed $\gamma_0 = \gamma/\sqrt{N}$. (a) We plot the test loss (MSE) and (b) test classification error. Networks with higher $\gamma_0$ train more rapidly. Time is measured in every 100 update steps. (c) The dynamics of the last layer feature kernel $\Phi^L$, shown as alignment to the target function. As predicted by the DMFT, higher $\gamma_0$ corresponds to more active kernel evolution, evidenced by larger change in the alignment.

Though our DMFT is quite general in regards to the data and architecture, the technique is not entirely rigorous and relies on heuristic physics techniques. Our theory holds in the $T, P = \mathcal{O}_N(1)$ and may break down otherwise; other asymptotic regimes (such as $P/N, T/\log(N) = \mathcal{O}_N(1)$, etc) may exhibit phenomena relevant to deep learning practice [32, 83]. The computational requirements of our method, while smaller than the exponential time complexity for exact solution [1], are still significant for large $PT$. In Table 1, we compare the time taken for various theories to compute the feature kernels throughout $T$ steps of gradient descent. For a width $N$ network, computation of each forward pass on all $P$ data points takes $\mathcal{O}(PN^2)$ computations. The static NTK requires computation of $\mathcal{O}(P^2)$ entries in the kernel which do not need to be recomputed. However, the DMFT requires matrix multiplications on $PT \times PT$ matrices giving a $\mathcal{O}(P^3T^3)$ time scaling. Future work could aim to improve the computational overhead of the algorithm, by considering data averaged theories [64] or one pass SGD [1]. Alternative projected versions of gradient descent have also enabled much better computational scaling in evaluation of the theoretical predictions [46], allowing evaluation on full CIFAR-10.

| Requirements | Width-$N$ NN | Static NTK | Perturbative | Full DMFT |
|---|---|---|---|---|
| Memory for Kernels | $\mathcal{O}(N^2)$ | $\mathcal{O}(P^2)$ | $\mathcal{O}(P^4T)$ | $\mathcal{O}(P^2T^2)$ |
| Time for Kernels | $\mathcal{O}(PN^2T)$ | $\mathcal{O}(P^2)$ | $\mathcal{O}(P^4T)$ | $\mathcal{O}(P^3T^3)$ |
| Time for Final Outputs | $\mathcal{O}(PN^2T)$ | $\mathcal{O}(P^3)$ | $\mathcal{O}(P^4)$ | $\mathcal{O}(P^3T^3)$ |

Table 1: Computational requirements to compute kernel dynamics and trained network predictions on $P$ points in a depth $N$ neural network on a grid of $T$ time points trained with $P$ data points for various theories. DMFT is faster and less memory intensive than a width $N$ network only if $N \gg PT$. It is more computationally efficient to compute full DMFT kernels than leading order perturbation theory when $T \ll \sqrt{P}$. The expensive scaling with both samples and time are the cost of a full-batch non-perturbative theory of gradient based feature learning dynamics.

## Acknowledgments and Disclosure of Funding

This work was supported by NSF grant DMS-2134157 and an award from the Harvard Data Science Initiative Competitive Research Fund. BB acknowledges additional support from the NSF-Simons Center for Mathematical and Statistical Analysis of Biology at Harvard (award #1764269) and the Harvard Q-Bio Initiative.

BB thanks Jacob Zavatone-Veth, Alex Atanasov, Abdulkadir Canatar, and Ben Ruben for comments on this manuscript as well as Greg Yang, Boris Hanin, Yasaman Bahri, and Jascha Sohl-Dickstein for useful discussions.

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
