# OpenReview forum: "Self-Consistent Dynamical Field Theory of Kernel Evolution in Wide Neural Networks"
_NeurIPS.cc/2022/Conference — NeurIPS 2022 Accept_

### Official Review · Reviewer_m42e · 2022-06-30

**Rating:** 7
**Confidence:** 4
**Soundness:** 3 good
**Presentation:** 3 good
**Contribution:** 3 good

**Summary:**

In this paper, the authors continues the line of research that uses dynamical mean field theory (DMFT) to study neural networks.
Using statistical physics heuristics, they derive self-consistent equations that governs the evolutions of the kernels, the (internal) representations and the gradients of the networks in the infinite-width feature learning regime for deep neural networks. Although these equations are too expensive to solve in the general setting, the authors propose a sampling-based approach to solve the equations numerically. They show good agreement between theory prediction and simulation. Finally, the authors have a very nice discussion explaining that their framework capture several recently developed perturbation-based approaches to understand neural networks.

Overall, I find the paper interesting, insightful and have valuable contribution to the NeurIPS community.

**Questions:**

- Is it possible to formulate the assumptions under which the results are theoretically sound?

- Yang&Hu (Feature Learning in Infinite-Width Neural Networks) also derives similar (recursive) equations that depend on the distribution of of the (hidden) representations and their derivatives. Followup work (EFFICIENT COMPUTATION OF DEEP NONLINEAR ∞-
WIDTH NEURAL NETWORKS THAT LEARN FEATURES) also derives efficient approximation scheme to solve the recursive equations, which is scalable to full CIFAR10. Can you add more detailed discussion about these two papers? What are the advantages and disadvantages? Is it also possible to scale up your algorithm to full CIFAR10?



**Limitations:**

The paper is purely theoretical. No potential negative societal impact as I can tell.

The authors discussed the limitation regarding computation. The non-rigorousness nature of the approach should also be discussed.

**Strengths And Weaknesses:**

## Strengths

- Generating several previous DFMT related works to multiple layer networks setting
- Technical contribution in deriving self-consistent equations
- Proposed framework captures several related existing works, e.g. finite-size correction [26, 27] in the paper.
- Good agreement between theory and (small scale) simulation.


## Weaknesses
- The calculations are very far from rigorous. It is totally unclear that under what assumptions the results of the paper are correct.
- The current presentation is not very friendly to readers without DMFT background. E.g. Section 3 (main theoretical contribution) is very hard to follow and hard to extract key ideas and insights behind the equations and the techniques for deriving them. Walking the readers through the deviation in the simplest possible setting (e.g. 2-layer linear networks) will be much appreciated.
- The self-consistent equations are not very interpretable (at least at the current form). It is not clear, at least to me, what extra insights regarding feature learning (beyond linear models) can we get from those equations.
-  Computationally, these equations are much much more expensive to solve than just training the original networks and unscalable (cubic dependence on both training steps and training samples)

---

> ### Author Response · Authors · 2022-08-01
> **Response Part 1**
>
>
> ### Strengths
>
> 1. *Generating several previous DFMT related works to multiple layer networks setting*
> 2. *Technical contribution in deriving self-consistent equations*
> 3. *Proposed framework captures several related existing works, e.g. finite-size correction [26, 27] in the paper.*
> 4. *Good agreement between theory and (small scale) simulation.*
>
> We thank the reviewer for their careful reading and review and for their appreciation of these aspects of our paper.
>
> ### Weaknesses
>
> #### Lack of Rigor and Conditions Under Which Theory Holds
> *The calculations are very far from rigorous. It is totally unclear that under what assumptions the results of the paper are correct... The non-rigorousness nature of the approach should also be discussed*
>
> We do not have a rigorous proof which starts from a collection of sufficient conditions and proceeds to prove the asymptotic validity of our derived DMFT equations. Rather our derivation relies on heuristics (a saddle point technique), which is commonly employed in statistical physics. Necessarily our method requires that the activation functions have a well defined second weak derivative for $L\geq 2$ (so that the $A,B$ order parameters are well-defined). Further our theory will be valid in a regime where $P,T \sim \mathcal{O}_N(1)$. Phenomena where the number of timesteps or samples scales with $N$ are currently inaccessible within the DMFT equations and require alternative techniques. It is possible, for instance, that effects where $T \sim \log N$ such as in this work (https://arxiv.org/abs/2202.04509), are not detectable in the DMFT limit. We add this stipulation on $T,P$ at the beginning of Section 3
> "Next, we derive our self-consistent DMFT in a limit where $t, P = \mathcal{O}_N(1)$"
>
>
> In our discussion, we add information about the limitations of this assumption and the lack of rigor in our approach:
> "Though our DMFT is quite general in regards to the data and architecture, the technique is not entirely rigorous and relies on heuristic physics techniques. Our theory holds in the $T,P = \mathcal{O}_N(1)$ and may break down otherwise. Other asymptotic regimes (such as $P/N, T/\log(N)=\mathcal{O}_N(1)$, etc) may exhibit phenomena relevant to deep learning practice."
>
>
> Finite size $N$ effects at fixed $P,T$ as well as other asymptotic regimes $P/ N = O_N(1)$ are worthy of future investigation. The rate of convergence with $N$ of the kernels to the DMFT predictions is also worth future investigation.
>
>
> #### Making a Simpler 2-Layer Derivation
> *The current presentation is not very friendly to readers without DMFT background. E.g. Section 3 (main theoretical contribution) is very hard to follow and hard to extract key ideas and insights behind the equations and the techniques for deriving them. Walking the readers through the deviation in the simplest possible setting (e.g. 2-layer linear networks) will be much appreciated.*
>
> We thank the reviewer for this great suggestion. We added a step-by-step derivation for the two layer case in the new Appendix D.2. We mention this new Appendix early in section 3.1: "For a simplified analysis of the $L=1$ case, see Appendix D.2"

---

> > ### Author Response · Authors · 2022-08-01
> > **Response Part 2**
> >
> >
> >
> > #### Interpretability of the Theory & Learned Features
> > We agree that at this stage the feature evolution equations are complicated nonlinear coupled integral equations which are not immediately interpretable.
> >
> > To improve interpretability of our DMFT stochastic process notation in Equation 10, we eliminated the step functions $\Theta(t-s)$ and instead now just integrate both terms for $s\in(0,t)$. This is legitimate since the response functions $A, B$ are causal.
> >
> > We think that developing more in depth interpretation of these equations, perhaps in special limits, could be useful in follow up works. However, we do want to defend the following insights which we think give some interpretation of our result:
> > 1. Each neuron's activation and gradient signal is an iid draw from a distribution defined by $\mathcal Z^\ell$.
> > 2. The updates to the pre-activations and pre-gradients are $O(\gamma_0)$. The $\gamma_0 \to 0$ limit is just a Gaussian, which recovers the static NTK picture where $\Phi$, $G$ can be computed at init and treated as constants through time.
> > 3. The feature learning updates are recursive nonlinear compositions of Gaussian random variables $u^\ell,r^\ell$.
> > 4. The preactivation $h^\ell$ update depends on the history of $\Phi^{\ell-1}$ while the pre-gradient update depends on the history of $G^{\ell+1}$ which intuitively shows that the $\Phi^\ell$ kernels accumulate corrections from first layer to last while $G$ kernels accumulate update from last layer to first. This is also visible from perturbation theory (see Appendix P)
> > 5. The $A$ and $B$ kernels quantify the sensitivity of feedforward signals to the feedback fields and vice versa.
> > 6. All dyamical updates depend on $\Delta_\mu = - \frac{\partial \mathcal L}{\partial f_\mu}$. Network predictions evolve according to the dynamical NTK $K = \sum_\ell G^{\ell+1} \Phi^\ell$. Empirically we see that the feature kernels tend to align to the target function $yy^\top$ which provides accelerates learning.
> >
> > #### Connection to Prior Works including Yang & Hu 2021
> >
> > We appreciate the reviewer's interest in the connection between our work and the work on the $\mu P$ limit of Yang & Hu. After more carefully reviewing Yang & Hu's derived stochastic process for fields, we find that the field equations we derived with our DMFT at $\gamma_0=1$ and discrete time agree with those derived by Yang & Hu with Tensor programs. We added several comments crediting Yang & Hu's work in this new draft of our manuscript.
> >
> > 1. In our abstract we now write
> > "We show that the field theory derivation recovers the recursive stochastic process of infinite-width feature learning networks obtained from Yang \& Hu with Tensor Programs \cite{yang2021tensor}."
> > 2. In the introduction we write
> > "Using the Tensor Programs framework, Yang \& Hu identified a stochastic process that describes the evolution of preactivation features in infinite-width $\mu P$ NNs \cite{yang2021tensor}. In this work, we study an equivalent parameterization to $\mu P$ with self-consistent dynamical mean field theory (DMFT) and recover the stochastic process description of infinite NNs using this alternative technique. In the same large width scaling, we include a scalar parameter $\gamma_0$ that allows smooth interpolation between lazy and rich behavior \cite{chizat2019lazy}. We provide a new computational procedure to sample this stochastic process and demonstrate its predictive power for wide NNs."
> > 3. In the Related Works section we added the following paragraph
> >
> > "Our results are most closely related to a set of recent works which studied infinite-width NNs trained with gradient descent (GD) using the Tensor Programs (TP) framework \cite{yang2021tensor}. We show that our discrete time field theory at unit feature learning strength $\gamma_0 = 1$ recovers the stochastic process which was derived from TP. The stochastic process derived from TP has provided insights into practical issues in NN training such as hyper-parameter search \cite{yang2021tuning}. Computing the exact infinite-width limit of GD has exponential time requirements \cite{yang2021tensor}, which we show can be circumvented with an alternating sampling procedure. A projected variant of GD training has provided an infinite-width theory that could be scaled to realistic datasets like CIFAR-10 \cite{yang2022efficient}. Inspired by Chizat and Bach's work on mechanisms of lazy and rich training \cite{chizat2019lazy}, our theory interpolates between lazy and rich behavior in the mean field limit for varying $\gamma_0$ and allows comparison of DMFT to perturbative analysis near small $\gamma_0$. Further, our derivation of a DMFT action allows the possibility of pursuing finite width effects."

---

> > > ### Author Response · Authors · 2022-08-01
> > > **Response Part 3**
> > >
> > >
> > > We also want to highlight some novel aspects of our work which expand on the original analysis of Yang & Hu. Concretely, these novel aspects are
> > >
> > > 1. Giving a novel derivation of the infinite width limiting stochastic process behavior in the mean field regime using techniques from statistical physics.
> > > 2. Provide a polynomial time/space numerical algorithm to solve the saddle point equations.
> > > 3. Allowing for any feature learning strength by including a richness parameter $\gamma_0$ and studying the accuracy of DMFT and various approximations across $\gamma_0$.
> > > 4. Performing a cursory analysis of finite size $N<\infty$ effects in Appendix P.7. Our DMFT action makes such computations technically straightforward.
> > > 5. Exploring the role of regularization (Appendix J) and Langevin noise (Appendix K), momentum (Appendix L).
> > > 6. Giving an exact one-dimensional dynamics of linear networks trained on whitened data for arbitrary $\gamma_0$ (Section 4.1 and Appendix F.1.1).

---

> ### Comment · Reviewer_m42e · 2022-08-03
> **Update**
>
> I appreciate the authors' efforts in making the following changes:
> - Including deviation of the simple $L=1$ case.
> - A more comprehensive comparison with Yang&Hu and follow-up work.
> - An acknowledgement that the derived formula is the same as in Yang&Hu. Note that the techniques used here are very different, which, in my opinion, is very a valuable contribution to the community.
> - Addressing several points raised in the review, e.g., the non-rigorousness nature of the approach.
>
> Overall, this is a very strong submission and I raise my score accordingly. Congrats to the authors!

---

### Official Review · Reviewer_mQQo · 2022-07-11

**Rating:** 7
**Confidence:** 3
**Soundness:** 3 good
**Presentation:** 3 good
**Contribution:** 3 good

**Summary:**

This work proposes to simulate the dynamics of infinitely-wide neural networks under gradient descent through a self-consistent dynamical field theory. The key idea is to adopt techniques from dynamical kernels, which can govern the evolution of a neural network. The authors provide an analytic result for the linear network while providing a sampling method to approximate the dynamics. Compared to various approximation sketches, the proposed method is shown to obtain consistent solutions across different regime. Lastly, the authors provide experiments in more realistic settings which demonstrate that the method is still valid.

**Questions:**

We know that NTK methods cannot fully explain the full capabilities of neural networks. What makes me curious is, what is the difference between the method in this work and the finite width neural network in practical performance, especially in the feature learning area?

**Limitations:**

I appreciate that the authors acknowledge the limitation of expensive computation. I encourage the authors to try to further improve the computational efficiency of the algorithm in future research.

**Strengths And Weaknesses:**

Overall, the work is of high quality. First of all, in terms of writing, this article has a clear structure and is relatively easy to read. From a method perspective, it is a novel idea to use dynamicsl field theory to simulate the dynamics of infinitely wide neural networks, especially the feature learning area. From the simulation results, compared with other basic methods, the method proposed by the author can effectively capture the dynamics of the neural network.

On the other hand, the method proposed in this work has a relatively large limitation in computational efficiency, and there is still a certain gap from the actual neural network application scenario. However, I also agree that this limitation can be addressed in the next step.

---

> ### Author Response · Authors · 2022-08-01
> **Response Part 1**
>
>
> ### Strengths
>
> *Overall, the work is of high quality. First of all, in terms of writing, this article has a clear structure and is relatively easy to read. From a method perspective, it is a novel idea to use dynamical field theory to simulate the dynamics of infinitely wide neural networks, especially the feature learning area. From the simulation results, compared with other basic methods, the method proposed by the author can effectively capture the dynamics of the neural network.*
>
> We thank the reviewer for their support.
>
>
> ### Weaknesses
>
> #### Computational Efficiency
> *On the other hand, the method proposed in this work has a relatively large limitation in computational efficiency, and there is still a certain gap from the actual neural network application scenario. However, I also agree that this limitation can be addressed in the next step.*
>
>
> Thank you for pointing this out. As we mention in our discussion, the cubic dependence on the samples and timesteps (solution requires $O(P^3 T^3)$ steps) imposes a strict limitation on the applicability of our solution method to realistic scale problems. We mention below some possible ways of improving this scaling in special cases.
>
> 1. Our theory can scale to arbitrary sample size $P$ for linear networks when the training data is whitened. For such networks and data, the DMFT can be solved entirely in terms of $T\times T$ matrices in $O(T^3)$ time. We provided the two layer ($L=1$) example in our original submission. In response to the reviewer comments, we provided the equations for deep $L \geq 2$ linear networks in the new Appendix F.2.
> 2. One can compute the *final* kernels and predictions in $O(P^3)$ time for regularized mean field training with Langevin noise (Gaussian white noise added to weights during training), using the equilibrium distribution. We added an analysis of Langevin training for mean field networks in Appendix K. The equilibrium analysis in Appendix K.3 gives a collection of equations for the kernels and final predictions which close.
> 3. Alternatives to exact gradient descent, including the projected $\pi$ gradient descent of Yang et al 2022 https://openreview.net/forum?id=tUMr0Iox8XW, have been shown to admit more efficient computations of the exact infinite width behavior. It would be interesting for future work to explore a DMFT derivation of these alternative efficient algorithms and to explore other possible alternatives to exact GD which give more efficient infinite width computations.
>
> There may be other ways we have not yet conceived of yet which give more practical computations of the infinite width feature learning setting.

---

> > ### Author Response · Authors · 2022-08-01
> > **Response Part 2**
> >
> >
> > #### Difference between NNs in this limit and practical finite width networks
> >
> > *We know that NTK methods cannot fully explain the full capabilities of neural networks. What makes me curious is, what is the difference between the method in this work and the finite width neural network in practical performance, especially in the feature learning area?*
> >
> > This is a very important question which we currently do not know the answer to. Our DMFT holds in a regime where $N \gg T, P$ and where each hidden layer is sufficiently wide for the kernels to become initialization-independent quantities. Realistic networks may not be in this regime.
> >
> > However, we attempted to provide some empirics which give insight into how well finite width $N$ networks are well approximated by our theory.
> >
> > 1. In Figure 1 (f) and (i) we show the cosine similarity between kernels predicted by DMFT and the empirical kernels of a width $N$ network for varying $N$. For this small problem in Figure 1, the kernel converges nearly perfectly to DMFT behavior after $N \sim 100$.
> >
> > 2. Further, in cases where the learning problem is too big for us to simulate our theory, we can still attempt to compare the behavior of the network at different $N$ for fixed $\gamma_0$. In Figure 4, we attempted to show that for $N \in \{250,500\}$ networks trained on 2 classes of CIFAR-10 the behavior of the network at different $N$ is almost identical if $\gamma_0$ is the same. A likely explanation is that, even at these modest widths, the network is already close to its DMFT behavior.  We are currently working on larger experimental sweeps over $N$ of this kind to visualize the convergence behavior of the kernel and loss dynamics, but have not yet finished these larger experiments. We will include them in the final version.
> >
> > 3. We also are trying to make progress on this question on the theory front. In the new Appendix P.7, we attempt to analyze leading order $\mathcal{O}( \frac{1}{N} )$ corrections to the dynamical kernels.  We find that the leading order finite size effects are variance inducing, ie lead to fluctuations in the kernels over $\theta_0$ (over the distribution of inits). We added a computation of the relevant components of the inverse correlation matrix for these fluctuations in Appendix P.7.1. If one expects that, at fixed $\gamma_0$, for $N \gg 1$ that the learned predictor can be modeled as $f = f_{N =\infty} + \delta f_N$ where $\delta f_N$ is a mean-zero stochastic process uncorrelated with the target function, then by the bias-variance decomposition the expected generalization MSE would decompose as $\left< \mathcal{L}_N \right> \approx \left< \mathcal{L}_{\infty} \right> + \left< (\delta f_N)^2 \right>$ , providing a prediction that finite size effects would increase the expected test loss of the model. This needs to be tested.

---

> > > ### Comment · Reviewer_mQQo · 2022-08-09
> > > **Update**
> > >
> > > Thank you for your detailed responses. I appreciate that the authors make an effect to further address my concerns. Thus I decide to keep my score and recommend acceptance.

---

### Official Review · Reviewer_DJ54 · 2022-07-12

**Rating:** 8
**Confidence:** 4
**Soundness:** 4 excellent
**Presentation:** 3 good
**Contribution:** 4 excellent

**Summary:**

This paper analyzes feature learning in infinite-width neural networks trained with gradient descent. In this limit, the distribution over hidden unit activations and gradients in each layer becomes i.i.d. and can be characterized with some order parameters using self-consistent dynamical field theory. These order parameters are inner-product kernels that are determined self-consistently from the aforementioned distributions. These self-consistent equations can be solved exactly for linear networks, as they reduce to matrix equations in this case, or numerically for nonlinear networks. The authors demonstrate very good agreement between their theory and simulations in finite-width neural networks. Finally, limitations of the theory are discussed, showing where the theory breaks down, and a comparison to other analyses of feature learning via perturbations of the NTK is made.


**Questions:**

- Do the authors have any sense how much of the performance gap between finite-width neural networks and infinite-width neural networks is closed by the feature learning in this limit?
- For linear networks, is the main obstacle to scaling the experiments to larger datasets in solving the linear system of Equation (12) or are there other challenges?

**Limitations:**

No concerns about negative societal impact.



**Strengths And Weaknesses:**

Strengths
- Understanding feature learning is arguably the most important open theoretical problem for neural networks and is certainly of interest and significance to the NeurIPS community.
- Obviously, the work builds on previous techniques and analyses but the paper makes a strong original contribution with interesting results.
- The discussion of connections to previous work are helpful for the reader to build intuition.

Weaknesses
- Most of the calculations are relegated to the supplement. This is probably expected given their complexity.
- It is not completely clear what is proved rigorously and what results are formal calculations.
- The experiments are on extremely small datasets due to the computational limitations discussed in Section 7.
- There is little interpretation of the features that are learned in this limit.

Minor typos
- Citation on line 198.
- "numerically method" on line 249.
- "Which" and "that" are mixed up several times.
- Some compound adjectives are missing hyphens, e.g. "infinite width neural networks" -> "infinite-width neural networks"
- "Figures 5 and Figure 6" on line 165

---

> ### Author Response · Authors · 2022-08-01
> **Response Part 1**
>
>
> ### Strengths
>
>
> 1. *Understanding feature learning is arguably the most important open theoretical problem for neural networks and is certainly of interest and significance to the NeurIPS community.*
> 2. *Obviously, the work builds on previous techniques and analyses but the paper makes a strong original contribution with interesting results.*
> 3. *The discussion of connections to previous work are helpful for the reader to build intuition.*
>
> We thank the reviewer for their careful reading and for appreciation of our motivation, results, and discussion.
>
> ### Weaknesses
>
> #### Calculations in Supplement
> *Most of the calculations are relegated to the supplement. This is probably expected given their complexity.*
>
> * Yes, most of our derivation is placed in the supplement due to space limitations. While this is not ideal, we wanted to focus the main text on the setup and figures displaying results.
>
> #### Rigorous vs Formal Results
> *It is not completely clear what is proved rigorously and what results are formal calculations.*
>
>
> Thank you for this comment. We do not have a rigorous proof which starts from a collection of sufficient conditions and proceeds to prove the asymptotic validity of our derived DMFT equations. Rather our derivation relies on heuristics (a saddle point technique), which is commonly employed in statistical physics. Necessarily our method requires that the activation functions have a well defined second weak derivative for $L\geq 2$ (so that the $A,B$ order parameters are well-defined). Further our theory will be valid in a regime where $P,T \sim \mathcal{O}_N(1)$. Phenomena where the number of timesteps or samples scales with $N$ are currently inaccessible within the DMFT equations and require alternative techniques. It is possible, for instance, that effects where $T \sim \log N$ such as in this work (https://arxiv.org/abs/2202.04509), are not detectable in the DMFT limit. We add this stipulation on $T,P$ at the beginning of Section 3
> "Next, we derive our self-consistent DMFT in a limit where $t, P = \mathcal{O}_N(1)$"
>
>
> In our discussion, we add information about the limitations of this assumption and the lack of rigor in our approach:
> "Though our DMFT is quite general in regards to the data and architecture, the technique is not entirely rigorous and relies on heuristic physics techniques. Our theory holds in the $T,P = \mathcal{O}_N(1)$ and may break down otherwise. Other asymptotic regimes (such as $P/N, T/\log(N)=\mathcal{O}_N(1)$, etc) may exhibit phenomena relevant to deep learning practice."
>
>
> Finite size $N$ effects at fixed $P,T$ as well as other asymptotic regimes $P/ N = O_N(1)$ are worthy of future investigation. The rate of convergence with $N$ of the kernels to the DMFT predictions is also worth future investigation.
>
>
> #### Small Datasets
> *The experiments are on extremely small datasets due to the computational limitations discussed in Section 7.*
>
>
> Thank you for bringing this up. As we mention in our discussion, the cubic dependence on the samples and timesteps (solution requires $O(P^3 T^3)$ steps) imposes a strict limitation on the applicability of our solution method to realistic scale problems. We mention below some possible ways of improving this scaling in special cases.
>
> 1. Our theory can scale to arbitrary sample size $P$ for linear networks when the training data is whitened. For such networks and data, the DMFT can be solved entirely in terms of $T\times T$ matrices in $O(T^3)$ time. We provided the two layer ($L=1$) example in our original submission. In response to the reviewer comments, we provided the equations for deep $L \geq 2$ linear networks in the new Appendix F.2.
> 2. One can compute the *final* kernels and predictions in $O(P^3)$ time for regularized mean field training with Langevin noise (Gaussian white noise added to weights during training), using the equilibrium distribution. We added an analysis of Langevin training for mean field networks in Appendix K. The equilibrium analysis in Appendix K.3 gives a collection of equations for the kernels and final predictions which close.
> 3. Alternatives to exact gradient descent, including the projected $\pi$ gradient descent of Yang et al 2022 https://openreview.net/forum?id=tUMr0Iox8XW, have been shown to admit more efficient computations of the exact infinite width behavior. It would be interesting for future work to explore a DMFT derivation of these alternative efficient algorithms and to explore other possible alternatives to exact GD which give more efficient infinite width computations.
>
> There may be other ways we have not yet conceived of yet which give more practical computations of the infinite width feature learning setting.

---

> > ### Author Response · Authors · 2022-08-01
> > **Response Part 2**
> >
> >
> >
> > #### Interpretation of the Features Learned in this Limit
> > *There is little interpretation of the features that are learned in this limit.*
> >
> > We agree that at this stage the feature evolution equations are complicated nonlinear coupled integral equations which are not immediately interpretable.
> >
> > To improve interpretability of our DMFT stochastic process notation in Equation 10, we eliminated the step functions $\Theta(t-s)$ and instead now just integrate both terms for $s\in(0,t)$. This is legitimate since the response functions $A, B$ are causal.
> >
> > We think that developing more in depth interpretation of these equations, perhaps in special limits, could be useful in follow up works. However, we do want to defend the following insights which we think give some interpretation of our result:
> > 1. Each neuron's activation and gradient signal is an iid draw from a distribution defined by $\mathcal Z^\ell$.
> > 2. The updates to the pre-activations and pre-gradients are $O(\gamma_0)$. The $\gamma_0 \to 0$ limit is just a Gaussian, which recovers the static NTK picture where $\Phi$, $G$ can be computed at init and treated as constants through time.
> > 3. The feature learning updates are recursive nonlinear compositions of Gaussian random variables $u^\ell,r^\ell$.
> > 4. The preactivation $h^\ell$ update depends on the history of $\Phi^{\ell-1}$ while the pre-gradient update depends on the history of $G^{\ell+1}$ which intuitively shows that the $\Phi^\ell$ kernels accumulate corrections from first layer to last while $G$ kenels accumulate update from last layer to first. This is also visible from perturbation theory (see Appendix P)
> > 5. The $A$ and $B$ kernels quantify the sensitivity of feedforward signals to the feedback fields and vice versa.
> > 6. All dyamical updates depend on $\Delta_\mu = - \frac{\partial \mathcal L}{\partial f_\mu}$. Network predictions evolve according to the dynamical NTK $K = \sum_\ell G^{\ell+1} \Phi^\ell$. Empirically we see that the feature kernels tend to align to the target function $yy^\top$ which provides accelerates learning.
> >
> > ### Questions
> >
> > #### Finite vs Infinite Network Behavior/Performance
> > *Do the authors have any sense how much of the performance gap between finite-width neural networks and infinite-width neural networks is closed by the feature learning in this limit?*
> >
> > This is a very important question which we currently do not know the answer to. Our DMFT holds in a regime where $N \gg T, P$ and where each hidden layer is sufficiently wide for the kernels to become initialization-independent quantities. Realistic networks may not be in this regime.
> >
> > However, we attempted to provide some empirics which give insight into how well finite width $N$ networks are well approximated by our theory.
> >
> > 1. In Figure 1 (f) and (i) we show the cosine similarity between kernels predicted by DMFT and the empirical kernels of a width $N$ network for varying $N$. For this small problem in Figure 1, the kernel converges nearly perfectly to DMFT behavior after $N \sim 100$.
> >
> > 2. Further, in cases where the learning problem is too big for us to simulate our theory, we can still attempt to compare the behavior of the network at different $N$ for fixed $\gamma_0$. In Figure 4, we attempted to show that for $N \in \{250,500\}$ networks trained on 2 classes of CIFAR-10 the behavior of the network at different $N$ is almost identical if $\gamma_0$ is the same. A likely explanation is that, even at these modest widths, the network is already close to its DMFT behavior.  We are currently working on larger experimental sweeps over $N$ of this kind to visualize the convergence behavior of the kernel and loss dynamics, but have not yet finished these larger experiments. We will include them in the final version.
> >
> > 3. We also are trying to make progress on this question on the theory front. In the new Appendix P.7, we attempt to analyze leading order $\mathcal{O}( \frac{1}{N} )$ corrections to the dynamical kernels.  We find that the leading order finite size effects are variance inducing, i.e. lead to fluctuations in the kernels over $\theta_0$ (over the distribution of inits). We added a computation of the relevant components of the inverse correlation matrix for these fluctuations in Appendix P.7.1. If one expects that, at fixed $\gamma_0$, for $N \gg 1$ that the learned predictor can be modeled as $f = f_{N =\infty} + \delta f_N$ where $\delta f_N$ is a mean-zero stochastic process uncorrelated with the target function, then by the bias-variance decomposition the expected generalization MSE would decompose as $\left< \mathcal{L}_N \right> \approx \left< \mathcal{L}_{\infty} \right> + \left< (\delta f_N)^2 \right>$, providing a prediction that finite size effects would increase the expected test loss of the model. This needs to be tested.

---

> > > ### Author Response · Authors · 2022-08-01
> > > **Response Part 3**
> > >
> > >
> > > #### Computational Obstacle for Linear Networks
> > > *For linear networks, is the main obstacle to scaling the experiments to larger datasets in solving the linear system of Equation (12) or are there other challenges?*
> > >
> > > * Yes, solving equation 12 is the only required computational step to analyze deep linear networks. If the data is whitened in deep linear networks, one can reduce the equations to a system on $T \times T$ matrices, since evolution of the features only occurs in a single direction in sample space (see last paragraph of 4.1 and the new Figure 7 for 2 layer example). The new Appendix F.2 discusses whitened data in the deep linear ($L\geq 2$) case.
> > >
> > > ### Typos
> > > *Citation on line 198.
> > > "numerically method" on line 249.
> > > "Which" and "that" are mixed up several times.
> > > Some compound adjectives are missing hyphens, e.g. "infinite width neural networks" -> "infinite-width neural networks"
> > > "Figures 5 and Figure 6" on line 165*
> > >
> > >
> > > * We thank the reviewer for pointing out these typos. They have been addressed.

---

> > > > ### Comment · Reviewer_DJ54 · 2022-08-08
> > > > **Response to authors**
> > > >
> > > > I'm grateful to the authors for their detailed response to all my questions. I still feel confident that the paper should be accepted and will keep my current score.

---

### Author Response · Authors · 2022-08-01
**Global Response: Summary of Changes Made**

We thank the reviewers for their careful reading, appreciation of our paper's strengths, and useful criticism about its weaknesses. Below we provide a list of the updates we made to the paper in response.

### Summary of Changes/Additions to Paper Since Last Draft

1. We added a two layer ($L=1$) warm up derivation in the new Appendix D.2 to provide a friendlier introduction to the method for non-DMFT experts. The two layer case is especially friendly since the only kernels needed are $\Phi,G$. Further the $\chi,\xi$ fields are static and Gaussian for $L=1$, which is not the case for $L\geq 2$.
2. We modified our discussion to mention, as an additional limitation, the lack of rigor of our approach and the restriction to non-extensive sample size and time: "Though our DMFT is quite general in regards to the data and architecture, the technique is not entirely rigorous and relies on heuristic physics techniques. Our theory holds in the $T,P = \mathcal{O}_N(1)$; other asymptotic regimes (such as $P/N, T/\log(N)=\mathcal{O}_N(1)$, etc) may exhibit phenomena relevant to deep learning practice." This last stipulation that neither $P$ nor $t$ be extensive in $N$ is now also mentioned at the beginning of section 3.
3. To improve interpretability of our DMFT stochastic process notation in Equation 10, we eliminated the step functions $\Theta(t-s)$ and instead now just integrate both terms for $s\in(0,t)$. This is legitimate since the response functions $A, B$ are causal.
4. We provided more detailed comparison between our work and the prior work on infinite width feature learning of Yang & Hu. We show that our DMFT technique gives the same stochastic process if we write our evolution equations in discrete time and take $\gamma_0=1$. This is interesting since two different techniques (DMFT and Tensor Programs) both give the same description of network training. We mention this connection in our new abstract, introduction, related works, and discussion, as well as the final sentence in Section 3.2. We also mention the difference between our polynomial-time algorithm for solving the self-consistent equations and the exact but exponential time algorithm of Yang & Hu in the new introduction contribution list, item 3, as well as in the discussion. The Appendix N shows that the parameterization we consider is equivalent to their $\mu P$ parameterization and the Appendix N.6 shows the equivalence of the feature evolution equations and provides a dictionary between our notation and theirs for the interested reader. We also mention Yang et al's follow up work $\pi$ limit, an alternative projected version of gradient descent which has an efficiently computable infinite width limit, in the related works and the discussion.
5. We have also added several extensions of our results to show the wide range practicability of the DMFT to commonly studied training methods

a. We provided theoretical expressions for the leading finite-width $N$ fluctuations in the kernels in the new Appendix P.7.1 by computing components of the Hessian of the DMFT action. This shows additional potential utility of the DMFT formalism, which can, in principle, access finite size effects to start closing the gap between infinite and realistic finite size networks.

b.  We added new extensions of our results about deep linear networks on whitened data (Appendix F.2), showing that solution only requires solving for $T \times T$ matrices. We added Figure 7 which verifies that solving the one-dimensional system reproduces accurate loss and kernel alignment dynamics in linear networks.

c. We provided an analysis of networks trained with weight decay (L2 regularization) in Appendix J. For homogenous networks, the final learned function is a kernel regression solution with the *final* NTK which can be determined from the field equations. The new Figure 8 shows that this theory is accurate and that the network asymptotes to non-zero train loss at large time when $\lambda > 0$. We make connections to the prior work of [Lewkowycz & Gur-Ari](https://arxiv.org/abs/2006.08643).

d. We analyzed Langevin trained NNs in the mean field limit (Appendix K) with both dynamical (Appendix K.1-2) and equilibrium (Appendix K.3) analyses. The dynamical analysis at large time (which is like studying $\lim_{t\to\infty} \lim_{N\to\infty} f(N,t)$ for observable $f$) allows all kernels to be written in terms of absolute time differences $\tau = |t-s|$, which could reduce the computational overhead. Further, the equilibrium analysis, allows one to solve for the *final* kernels with only $\mathcal{O}(P^3)$ time complexity and has a [Bayesian interpretation](https://arxiv.org/abs/2108.13097). This limit is like studying the behavior of $\lim_{N\to\infty } \lim_{t \to \infty} f(N,t)$. We think that the in-depth similarities and differences between these two limits, as well as the fluctuation dissipation relationships in DMFT are worthy future study.

---

### Meta-Review · Area_Chair_ry1x · 2022-08-28

**Recommendation:** Accept
**Confidence:** Certain

**Metareview:**

This paper analyzes a dynamical mean field theory that describes feature learning via gradient flow for certain infinite-width neural networks. Self-consistent equations for the order parameters characterizing the dynamics are presented and methods for approximate numerical evaluation are discussed. Overall, this is a solid paper that advances the theory and understanding of feature learning for neural networks of large width and the reviewers and I unanimously support acceptance.



**Award:**

No

---

### Decision · Program_Chairs · 2022-09-14

Accept